# Inhibition of NAMPT by PAK4 Inhibitors

**DOI:** 10.3390/ijms251810138

**Published:** 2024-09-21

**Authors:** Yiling Wang, Audrey Minden

**Affiliations:** Susan Lehman Cullman Laboratory for Cancer Research, Department of Chemical Biology, Ernest Mario School of Pharmacy, Rutgers, The State University of New Jersey, Piscataway, NJ 08854, USA; yw798@scarletmail.rutgers.edu

**Keywords:** dual inhibitors, PAK4, PAK1, NAMPT, NAD, cancer treatment

## Abstract

The serine/threonine kinase PAK4 plays a crucial role in regulating cell proliferation, survival, migration, and invasion. Overexpression of PAK4 correlates with poor prognosis in some cancers. KPT-9274, a PAK4 inhibitor, significantly reduces the growth of triple-negative breast cancer cells and mammary tumors in mouse models, and it also inhibits the growth of several other types of cancer cells. Interestingly, although it was first identified as a PAK4 inhibitor, KPT-9274 was also found to inhibit the enzyme NAMPT (nicotinamide phosphoribosyltransferase), which is crucial for NAD (nicotinamide adenine dinucleotide) synthesis and vital for cellular energy and growth. These results made us question whether growth inhibition in response to KPT-9274 was due to PAK4 inhibition, NAMPT inhibition, or both. To address this, we tested several other PAK4 inhibitors that also inhibit cell growth, to determine whether they also inhibit NAMPT activity. Our findings confirm that multiple PAK4 inhibitors also inhibit NAMPT activity. This was assessed both in cell-free assays and in a breast cancer cell line. Molecular docking studies were also used to help us better understand the mechanism by which PAK4 inhibitors block PAK4 and NAMPT activity, and we identified specific residues on the PAK4 inhibitors that interact with NAMPT and PAK4. Our results suggest that PAK4 inhibitors may have a more complex mechanism of action than previously understood, necessitating further exploration of how they influence cancer cell growth.

## 1. Introduction

The p21-activated kinase (PAK) family is a family of serine/threonine kinases, originally identified as effectors of the GTPases Rac1 and Cdc42 [1,2]. Based on their structures and sequences, PAKs are classified into two groups: group I consists of PAKs 1–3 and group II consists of PAKs 4–6 [3,4,5]. Both group I and group II PAKs contain a C-terminal catalytic kinase domain and an N-terminal regulatory domain that includes the GTPase-binding domain (GBD) [5]. The PAKs bind to Cdc42 or Rac via the GBD domain [3,5,6,7,8]. The major differences between group I and II PAKs lie in the regulatory domains, although their kinase domains also have some differences in sequence. PAK family members play pivotal roles in regulating various cellular functions, including cell proliferation, cytoskeletal organization, motility, cell survival, and apoptosis [9,10,11,12,13].

PAK4, a member of the group II PAKs, has important roles in cell growth, survival, and motility [14,15]. PAK4 has been implicated in some types of cancer due to its role in regulating cell growth and it is overexpressed in many types of cancer cells, including breast cancer, kidney cancer, pancreatic cancer, ovarian cancer, and colorectal cancer [16,17,18,19]. Overexpressed PAK4 enhanced proliferation, migration, and invasion in MDA-MB-231 breast cancer cells by activating the PI3K/AKT pathway [20]. RNAi knockdown of PAK4 increased tumor-suppressive miRNAs that regulate apoptosis and cell survival, and also enhanced apoptosis by disrupting PAK4’s phosphorylation of Bad in pancreatic cancer cells [21,22,23]. PAK4 overexpression was associated with shorter survival and poor prognosis in some cancers, such as non-small cell lung cancer, ovarian cancer, and breast cancer [4,20,24]. Tissue samples from breast cancer patients with higher PAK4 levels displayed larger tumors, lymph node metastasis, advanced cancer stages, and poorer survival outcomes, compared to those with lower PAK4 levels [20]. PAK4 knockdown in non-small cell lung cancer cells reduced LIMK1 phosphorylation, decreasing cell migration and invasion [24]. The results from these studies suggest that PAK4 may be a promising target for cancer treatment. PAK4 inhibitors, such as KPT-9274 and PF-03758309, have been shown to reduce tumor growth in various cancer cell lines and in animal studies [4,16,25,26]. KPT-9274 induced apoptosis and inhibited invasion and migration in kidney cancer cells [26]. KPT-9274, combined with chemotherapy drugs, exhibited antitumor activity by inhibiting proliferation in mouse models of pancreatic ductal adenocarcinoma (PDAC) and inducing apoptosis in PDAC cancer cells [27]. Meanwhile, the inhibition of PAK4 by KPT-9274 may also modulate the immune system by reducing PD-L1 expression, potentially enhancing antitumor immune responses [28]. Our previous research demonstrated that KPT-9274 reduced the viability of different breast cancer cell lines, particularly three different TNBC cell lines. This was the case not only in cell lines, but also in animal studies. In mouse models of TNBC, tumorigenesis was reduced following oral administration of KPT-9274 [17]. It is exciting that KPT-9274 can inhibit the growth of TNBC, because TNBC is a type of breast cancer that is especially difficult to treat [16,29].

Interestingly, while KPT-9274 was initially identified as a PAK4 inhibitor, it was subsequently found to inhibit both PAK4 and NAMPT, making it a dual-specific inhibitor [30]. NAMPT is also an enzyme with important roles in cell growth. NAMPT is a key enzyme in the salvage pathway for NAD+ biosynthesis [30,31,32,33,34,35]. This pathway is the predominant pathway for NAD synthesis in mammalian cells, although NAD can also be made via two other pathways [31,32,36]. NAD is converted into NADPH via the pentose phosphate pathway, and changes in NAD+ levels directly affect NADP+ levels [35,37]. NAD+ is essential for energy production, DNA repair, and immune function, and its depletion is harmful to cells. NAD is a critical cofactor for enzymes in glycolysis, the citric acid cycle, and oxidative phosphorylation. These metabolic pathways provide energy needed for cell growth and proliferation. By regulating NAD levels, NAMPT plays a key role in sustaining the high energy demands of proliferating cells. This is especially important in cancer cells, which proliferate at a high level and have high energy levels. NAMPT was found to be improperly activated in some types of cancer. Some NAMPT inhibitors, such as FK866, CHS828, GEN618, and OT-82, have been shown to significantly inhibit cell growth in both in vitro and in vivo models of various cancers, including pancreatic cancer, non-small cell lung cancer, lymphoma, prostate cancer, and breast cancer [30,36,38,39]. CHS828 has shown cytotoxic effects in multiple myeloma cell lines, such as RPMI-8226, by reducing cell proliferation and inducing apoptosis [35,38]. OT-82 induced cell death through NAMPT inhibition in vitro and in mouse xenograft models of hematopoietic malignancies, and it was used in phase I trials for relapsed or refractory lymphoma (NCT03921879) [40]. The NAMPT inhibitor, FK866, was evaluated in various nude mouse xenograft models (e.g., ovarian cancer, pancreatic cancer, and colon cancer), and showed strong efficacy in phase I and II clinical trials for advanced or metastatic solid tumors, refractory lymphoma, and advanced pancreatic cancer [30,36,41]. FK866 inhibits NAMPT, disrupting NAD+ production and impairing energy metabolism, leading to apoptosis in MiaPaCa-2 pancreatic cancer cells [31,42,43]. In multiple myeloma cells, FK866 also increased reactive oxygen species (ROS) levels, disrupted the metabolism, and reduced ATP production, resulting in cell death [31,42,43,44,45]. GNE-618, induced the death of A549 non–small cell lung carcinoma cells and dramatically reduced tumor growth in a patient-derived tumor xenograft model [42]. These studies suggest that NAMPT may also be a promising target for inhibiting the growth of cancer cells.

The finding that KPT-9274 can inhibit not only PAK4, but also NAMPT, raises questions about the mechanism by which it can inhibit cell growth. In particular, it raises the question of whether it inhibits cancer cell growth via PAK4 inhibition, NAMPT inhibition, or both. In addition to KPT-9274, several other PAK4 inhibitors have also been shown to have growth-inhibitory functions in cancer cells, as described above. One way to begin to address the question of how PAK4 inhibitors reduce cell growth is to determine whether other PAK4 inhibitors also inhibit NAMPT. If so, this may suggest that NAMPT inhibition may be an important mechanism by which PAK4 inhibitors regulate cell growth, rather than PAK4 inhibition alone. To test this, we performed NAMPT activity assays in response to several PAK4 inhibitors. For comparison, we also tested several group I PAK inhibitors and several other controls, such as MEK inhibitors. We tested the effects of these inhibitors on NAMPT enzymatic activity in cell-free assays and the levels of NAD in the cells. Interestingly, we found that all PAK4 inhibitors inhibited NAMPT activity. Computational molecular docking was used to analyze the interactions between the inhibitors and PAK4 or NAMPT to enhance our understanding of their mechanism. The simulations predicted that PAK4 inhibitors bind to both NAMPT and PAK4, due to their pyrimidine and benzo derivatives, potentially affecting the efficacy of NAMPT inhibition.

## 2. Results

### 2.1. PAK4 Inhibitors Inhibit NAMPT Activity

We tested various PAK inhibitors in in vitro cell-free assays to determine whether they inhibit NAMPT activity (see Figure 1). In addition to KPT-9274, we also tested PF-3758309 (a PAK4 inhibitor), FRAX486 (which primarily targets group I PAKs with weak PAK4 inhibition), IPA-3 (a group I PAK inhibitor), and the newer PAK4 inhibitors, GNE-2861 and LCH-7749944. Notably, the PAK4 inhibitors KPT-9274 and PF-3758309 demonstrated significant inhibition of NAMPT, suggesting their dual-specific nature. PF-3758309 reduced NAMPT to 15.8%, compared to the control. (Figure 1A,B). The other PAK4 inhibitors (GNE-2861 and LCH-7749944) also showed significant NAMPT inhibition compared to the control, similar to KPT-9274. For the four PAK4 inhibitors, NAMPT activity ranged from 15.8% to 35.7% of the control, whereas for the two group I PAK inhibitors (FRAX486 and IPA-3), NAMPT activity ranged from 50.0% to 61.9% of the control. All negative controls, including phloretin (a natural compound unrelated to PAK signaling) [46,47,48], BI-847325 (an ATP-competitive MEK inhibitor) [49], and PD-0325901 (an MEK inhibitor with non-ATP-competitive properties) [50], showed no NAMPT inhibition. The well-known NAMPT inhibitor, FK-866, the positive control, exhibited strong NAMPT inhibition. The inhibition efficiency was observed over time, with a notable decrease in NAMPT activity at 30 min for all PAK4 inhibitors (Figure 1C,D). Overall, these findings suggest a relationship between PAK4 inhibitors and NAMPT inhibition, particularly among group II PAK inhibitors.

### 2.2. PAK4 Inhibitors Reduce NAD/NADH Levels in Cells

To further validate the effects of the inhibitors on NAMPT activity, we also measured NAD/NADH production in SUM159 cells, which should generally reflect NAMPT activity (see Figure 2). The inhibitors included KPT-9274, PF-3758309, FRAX486, and IPA-3. After 6 h of treatment, all PAK inhibitors resulted in a significant decrease in NAD/NADPH levels, as did the NAMPT inhibitor FK866 (Figure 2A). By 24 h, in an independent experiment in which two different doses of each inhibitor were tested, all the PAK inhibitors continued to lead to decreased NAD levels. The PAK4 inhibitors (KPT-9274, PF-3758309) were more effective than the other PAK inhibitors (Figure 2B). FRAX-486, which preferentially inhibits group I PAKs (PAK1-3), reduced NAD levels at day 1 (See Figure 2B). Similar results were seen for IPA-3, which also reduced NAD levels at day 1 (See Figure 2B). For the control group, the NAD level of phloretin was very close to that of the untreated group at 6 h, even higher than the untreated group by day 1. Our results indicated that PAK4 inhibitors reduced NAD/NADH levels in TNBC cells (SUM159). The reduction in NAD/NADH levels by PAK4 inhibitors was greater than by group I PAK1-3 inhibitors.

### 2.3. PAK4 Inhibitors Reduce NADP/NADPH Levels in Cells

Since NAD+ levels impact NADP+ levels, inhibiting NAMPT is expected to reduce NADP/NADPH levels. We measured NADP+ levels in SUM159 cells treated with various inhibitors, including two additional PAK4 inhibitors, GNE-2861 and LCH-7749944 (see Figure 3). Like the NAD/NADH assay, the PAK4 inhibitors (KPT-9274 and PF-3758309) significantly reduced the NADP/NADPH levels by 24 h. Consistent with the NAD/NAPD assays above, all PAK4 inhibitors exhibited lower NADP/NADPH levels than the group I PAK inhibitors, FRAX-486 and IPA3. Our results indicated that PAK4 inhibitors reduced NADP/NADPH levels in TNBC cells (SUM159). PAK4 inhibitors reduced the NADP/NADPH levels more effectively than group I PAK1-3 inhibitors after 24 h.

### 2.4. Computational Molecular Docking Predicted That PAK4 Inhibitors Had More Interactions with NAMPT Compared to Group I PAK Inhibitors

Molecular docking simulations using the MOE software (2022 version) were used to investigate how the different compounds inhibit PAK4 and NAMPT, by analyzing their interactions and binding affinities. The results indicated that PAK4 inhibitors interact with NAMPT via pyrimidines and their benzo derivatives on the inhibitors (see Figure 4). KPT-9274, bearing a pyridine core, is predicted to bind to both the PAK4 and NAMPT protein (Figure 4A). PF-3758309 bears an imidazole and a pyrazole and is predicted to bind to PAK4 and NAMPT through these groups, respectively (Figure 4B). GNE-2861 and LCH-7749944 also had groups containing pyrimidines and benzo derivatives, which were predicted to bind to NAMPT (Figure 4C). FRAX486 is a group I inhibitor, but it leads to partial PAK4 inhibition. It contains an imidazole moiety that is predicted to bind to both PAK4 and NAMPT (Figure 4D). Conversely, PAK1 inhibitors (IPA-3) and the control group (phloretin) lacked pyrimidines and their benzo derivatives (Figure 4E,I). The details of the docking results are provided in Table 1 and Table 2, and Figure 5 and Figure 6. Overall, our molecular docking simulations suggest that PAK4 inhibitors have a strong binding affinity with NAMPT and potentially interact with NAMPT more efficiently, compared to group I PAK1-3 inhibitors.

## 3. Discussion

To determine the relationship between PAK4 inhibitors and NAMPT inhibition, we investigated NAMPT activity in cell-free assays in response to six different PAK inhibitors (KPT9274, PF-3758309, FRAX486, IPA-3, GNE-2861, and LCH-7749944). The results showed that all PAK4 inhibitors (PF-3758309, GNE-2861, and LCH-7749944) reduced the NAMPT activity in cell-free assays. In addition to the cell-free assays, we assessed the NAMPT activity level in cells. One way to measure NAMPT activity is to measure the level of NAD+ in cells. This is not as precise as a cell-free assay because multiple factors may contribute to NAD+ levels in cells, but it can give us a window into what is happening in cells upon treatment with inhibitors [36]. We found that the PAK4 inhibitors reduced the levels of NAD/NADH at 6 h in SUM159 cells. The group I PAK inhibitors also inhibited NAD/NADH levels, although to a lower extent. We also tested various concentrations of each drug at 24 h. Our assay is complicated because NAD+ is essential for cell growth [51] and, consequently, some of the cells did not survive the drug treatment.

NADP+ is the phosphorylated form of NAD+ [41] and can also be used as a measurement of NAMPT activity. Therefore, we also assessed NADP+ levels in SUM159 cells. Similar to the NAD+ assay, PAK4 inhibitors reduced the NADP/NADPH levels at both 6 h and 24 h. The NADP+ levels in cells treated with PAK inhibitors decreased further with increasing concentrations of inhibitors. The exception is PF-3758309. Although PF-3758309 inhibited NADP+ production, the inhibition was slightly less at a higher concentration of the inhibitor. It is unclear why this is the case, but as noted above, using cells complicates the results, and compensatory mechanisms may occur in the cells over time. Therefore, it is important to consider these results alongside our cell-free assay, which directly measures NAMPT activity, as well as our cell-based NAD assay.

We conducted modeling studies using molecular docking software to help us better understand the mechanisms by which the drugs interact with their target proteins (NAMPT and PAK4). We found the following results: (1) pyrimidine and benzo derivatives in PAK4 inhibitors (KPT9274, PF-3758309, GNE-2861, LCH-7749944, and FRAX486) are predicted to interact with NAMPT or PAK4, showing stronger affinity compared to PAK1-3 inhibitors and controls. Among the various PAK inhibitors, group II PAK4 inhibitors (except LCH-7749944) exhibited a better binding score (−7.2~−7.8) with NAMPT than group I PAK inhibitors (FRAX486 and IPA-3), which had binding scores of −6.4 to −6.66, potentially leading to less efficient inhibition of NAMPT. A more negative binding affinity indicates a stronger ligand–receptor interaction [52]. (2) Hydrogen bonds appeared to be the major interactions between drugs and their target proteins, influencing molecular dynamics, enzymatic activity, and protein–ligand recognition [53]. Among all the PAK inhibitors, KPT-9274 had more interactions with NAMPT through stronger hydrogen bonds compared to FRAX486, potentially enhancing NAMPT inhibition. IPA-3, a PAK1 inhibitor, was predicted to inhibit NAMPT with weaker affinity, likely due to its lack of pyrimidine groups. (3) All the PAK inhibitors contained pi bonds, except GNE2861 and LCH-7749944, which enhanced their predicted interactions with NAMPT, even though their binding energies were often weaker than hydrogen bonds. (4) The position of nitrogen on the benzene ring influenced the electrostatic interactions, leading to NAMPT inhibition. The π system of the aromatic ring induced conformational changes for protein folding [53,54]. Comparing KPT9274 and PF-3758309 docking with NAMPT, differences in the hydrogen bond distances and energies are observed. NH1 in ARG40(B) interacted with N9 in KPT9274’s benzene ring, while N4 in PF-3758309’s benzene ring interacted with GLU202(A), with energy values of 3.4 kcal/mol and 1.5 kcal/mol, respectively.

We also tested the negative control, phloretin, in the MOE model. We confirmed that, as expected, this drug showed no interactions with NAMPT and lacked common chemical groups, such as pyrimidine. The results using phloretin confirmed our assay results, because it did not inhibit NAMPT or reduce NAD+ or NADP+ levels in cells. The results for the MEK inhibitors were more confusing because although MEK inhibitors showed interactions in virtual models, there was no inhibition in the NAMPT activity assays. It should be noted, however, that molecular modeling is a computer model. It can be used for prediction, but also has some limitations, due to being a computer modeling system. Cellular and even cell-free environments can be more complex than computational models due to factors like solubility, pharmacokinetics, metabolic stability, cellular barriers, and cell signaling networks. Despite its limitations, computational molecular modeling is highly beneficial for virtual drug evaluation, especially in large-scale databases. Future optimization and the inclusion of more cellular or tumor factors can enhance its effectiveness. Our work with molecular modeling can help us understand how PAK inhibitors may interact with NAMPT or PAK4 and can potentially help identify new dual PAK–NAMPT inhibitors.

Our results indicate that PAK4 inhibitors can inhibit not only PAK4, but also NAMPT. Our cell-free assays provide evidence that all the inhibitors can directly block NAMPT. Several studies have also demonstrated direct inhibition of PAK4 by PAK4 inhibitors in cell-free assays [55,56,57]. This is consistent with our MOE results, showing that PAK4 inhibitors directly interact with NAMPT and with PAK4. However, we cannot completely rule out the possibility that the drugs may also inhibit their targets indirectly to some extent.

Understanding how PAK4 inhibitors operate is important, as they have been shown to reduce the growth of several types of cancer cells, highlighting their potential for cancer therapy. Our studies raise interesting new questions about the mechanism by which these inhibitors suppress cancer cell growth. Although KPT-9274 was previously shown to inhibit both PAK4 and NAMPT, we had assumed that most PAK4 inhibitors block cancer cell growth primarily due to the inhibition of PAK4. However, now we find that all the PAK4 inhibitors we tested also inhibit NAMPT, causing us to reconsider the role of NAMPT in terms of their growth inhibitory properties. Our results raise the question as to whether the growth suppression properties of PAK4 inhibitors may in fact be mediated by NAMPT inhibition rather than PAK4 inhibition. Alternatively, both NAMPT and PAK4 inhibition may be required for growth inhibition, and the dual inhibition may even lead to a synergistic effect on cell growth. In the future, developing a better understanding about the molecular mechanism by which these inhibitors regulate cell growth will be essential for the development of the most effective strategies for controlling the growth of cancer cells.

## 4. Materials and Methods

### 4.1. Cell Lines and Reagents

KPT-9274 from Karyopharm Therapeutics Inc. (Newton, MA, USA) was dissolved in dimethyl sulfoxide (DMSO) to a final concentration of 10 mM. PF-3758309, FRAX486, IPA-3, GNE-2861, LCH-7749944, BI-847325, mirdametinib (PD0325901), and phloretin were purchased from MedChemExpress LLC (Monmouth Junction, NJ, USA) and were dissolved in DMSO. All the drugs were stored at −20 °C, until they were used. DMSO was stored at room temperature. SUM159 cells (from Asterland) were maintained in Ham’s F12 medium, supplemented with 5% FBS.

### 4.2. NAMPT Activity Cell-Free Assay

To detect the effect of PAK inhibitors (KPT-9274, PF-3758309, FRAX486, IPA-3, GNE-2861, and LCH-7749944) on NAMPT activity, recombinant NAMPT activity was measured using an enzyme-coupled reaction (CycLex NAMPT Colorimetric Assay Kit cat # CY-1251: CycLex Co., Ltd. (Nagano, Japan)). For screening inhibitors, the two-step assay method was used, following the manufacturer’s instructions. First, the Two-Step Assay Mixture-I and Two-Step Assay Mixture-II were prepared within 30 min before use. The Two-Step Assay Mixture-I consists of an NAMPT assay buffer, ATP, nicotinamide nucleotide adenylyltransferase 1 (NMNAT1), nicotinamide, and phosphoribosyl pyrophosphate (PRPP). The Two-Step Assay Mixture-II consists of water-soluble tetrazolium salts (WST-1), alcohol dehydrogenase (ADH), diaphorase, and ethanol. NAMPT was incubated with PAK inhibitors or control drugs in the presence of the Two-Step Assay Mixture-I and incubated at 30 °C for 60 min. The Two-Step Assay Mixture-II was then added to each sample and incubated at 30 °C. The absorbance of the samples and control drugs was detected at 450 nm at 0 min and 30 min, using a plate reader (Infinite 200 PRO, Tecan Austria GmbH, Grödig, Austria).

### 4.3. NAD/NADH Assay

The NAD/NADH-Glo assay (catalog no. G9071) kit was obtained from Promega (Madison, WI, USA). The assays were conducted in triplicate, according to the manufacturer’s protocol. The cells were plated at 5000 cells/well in white-walled, 96-well plates and incubated overnight. The old media were replaced by fresh media containing the different inhibitors. After 24 h of treatment, the NAD/NADH-Glo Detection Reagent was added into each well and incubated for 60 min at room temperature. This reagent contains a formulation that reacts with NADH to produce a luminescent signal. After cell lysis, the total NAD/NADH was assessed by luminescence using a plate reader (Infinite 200 PRO, Tecan Austria GmbH). The NAD+ levels are measured by converting NAD+ into NADH, which then reduces a substrate to luciferin. The light emitted by luciferin, detected by Ultra-Glo™ rLuciferase, reflects the levels of NAD+ and NADH in the sample.

### 4.4. NADP/NADPH Assay

The NADP/NADPH-Glo assay (catalog no. G9081) kit was obtained from Promega. This kit works according to a similar mechanism, but NADP/NADPY is measured instead of NAD/NADH. The assays were conducted in triplicate and according to the manufacturer’s protocol. The cells were plated at 5000 cells/well in white-walled, 96-well plates and incubated overnight. The old media were removed and replaced by fresh media containing different inhibitors. The NADP/NADPH-Glo Detection Reagent was added into each well at 6 h and 24 h treatment. The reaction was incubated for 60 min at room temperature. The cells were lysed and the total NADP/NADPH was measured by luminescence using a plate reader (Infinite 200 PRO, Tecan Austria GmbH). The NADP+ levels are measured by converting NADP+ into NADPH, which then reduces a substrate to luciferin. The light emitted by luciferin, detected by Ultra-Glo™ rLuciferase, reflects the levels of NADP+ and NADPH in the sample.

### 4.5. Molecular Docking Model

The MOE software’s (Molecular Operating Environment, 2022 version, accessed on 30 August 2022) docking modules were used to perform docking-based virtual analyses on the targeted NAMPT protein or PAK4 protein. The protein data bank (www.rcsb.org) was used to derive the crystal structure of the human NAMPT protein (PDB: 5U2N) and PAK4 protein (PDB: 4FIE). Compounds were docked into a protein’s active pocket. For visualization and posture creation, the compound with high-binding energy was chosen. The 2D and 3D interactions of the docked conformation were visualized using an inbuilt visualizer tool in MOE [58].

### 4.6. Statistical Analysis

All data were presented as mean ± standard deviation (STD), presented as error bars. An independent Student’s *t*-test was used to compare the means of the two groups. Any two groups referred to as significantly different in the text had a *p* < 0.05. All the statistical analysis was conducted by using SPSS software (Version 17.0, SPSS, Inc., Chicago, IL, USA).

## 5. Conclusions

In summary, our investigation revealed that several PAK4 inhibitors not only inhibit PAK4 activity, but also NAMPT activity, and lead to reduced NAD and NADP levels in SUM159 TNBC cells. Molecular docking studies support these findings, showing predicted interactions between PAK4 inhibitors and both PAK4 and NAMPT. PAK4 inhibitors are considered to be promising targets for cancer therapy. Our work raises important questions about the link between PAK4 and NAMPT inhibition in cell growth, emphasizing the need for further research to identify the most effective inhibitors in cancer treatment.

## Figures and Tables

**Figure 1 ijms-25-10138-f001:**
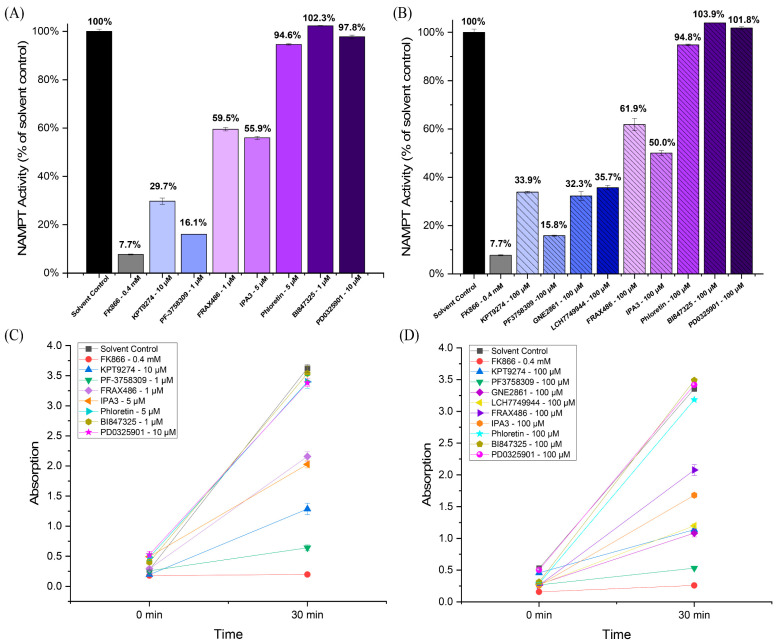
PAK4 inhibitors inhibit NAMPT activity in a cell-free assay. (**A**,**B**) NAMPT activity in response to PAK4 inhibitors (KPT-9274, PF-3758309, GNE2861, and LCH7749944), group I (PAK1-3) inhibitors (IPA-3 and FRAX486), phloretin (a natural compound unrelated to kinase inhibitors), MEK inhibitors (BI-847325 and PD0325901), the solvent control (DMSO), and FX866, the positive control. (**C**,**D**) Absorption (A450), reflecting NAMPT activity for PAK4 inhibitors at 0 min and 30 min. The results indicate that all PAK4 inhibitors inhibited NAMPT.

**Figure 2 ijms-25-10138-f002:**
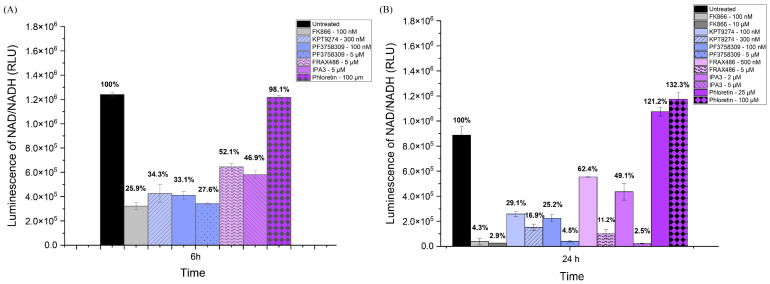
PAK4 inhibitors reduced the NAD/NADH levels in SUM159 cells. NAD/NADH assays of SUM159 cells treated with PAK4 inhibitors at 6 h (**A**) and 24 h (**B**). The individual values on the bar graph are the percentage of NAD/NADH compared to the control group. The results indicate that NAD/NADH levels significantly decreased in response to all the PAK4 inhibitors at both 6 h and 24 h.

**Figure 3 ijms-25-10138-f003:**
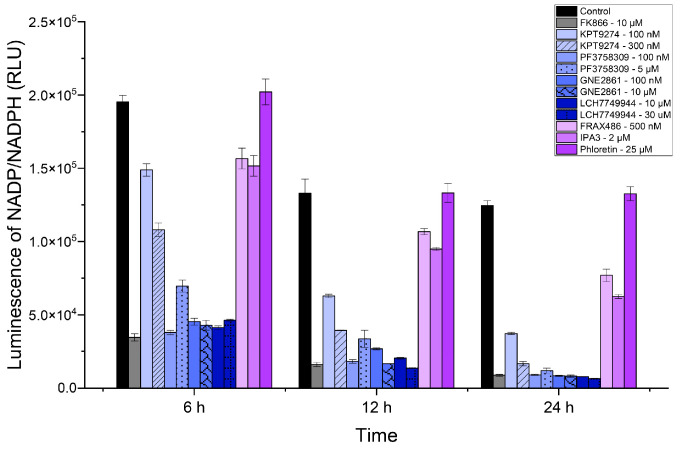
PAK4 inhibitors reduced NADP/NADPH levels in SUM159 cells. Luminescence corresponding to NADP/NADPH (RLU) levels in SUM159 cells treated with PAK inhibitors at 6 h, 12 h, and 24 h.

**Figure 4 ijms-25-10138-f004:**
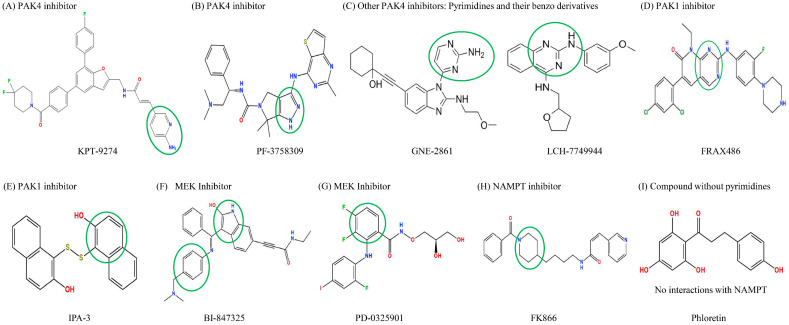
Structure of PAK inhibitors and control drugs. (**A**). KPT-9274, bearing a pyridine core. (**B**) PF-3758309, bearing an imidazole and a pyrazole. (**C**) Other PAK4 inhibitors (GNE-2861 and LCH-7749944) bearing pyrimidines and their benzo derivative. (**D**) FRAX486, a group I PAK inhibitor that also partially inhibits PAK4, bears an imidazole. (**E**) PAK1 inhibitor IPA-3; (**F**,**G**) MEK inhibitors BI-847325 and PD0325901; (**H**) NAMPT inhibitor FK866; (**I**) Phloretin, a compound without imidazole and pyrazole, unlike PAK4 inhibitors. Based on the chemical structures, the pyrimidines or benzo derivatives in PAK4 inhibitors are predicted to bind to NAMPT or PAK4. The groups that interact with NAMPT are highlighted with green circles.

**Figure 5 ijms-25-10138-f005:**
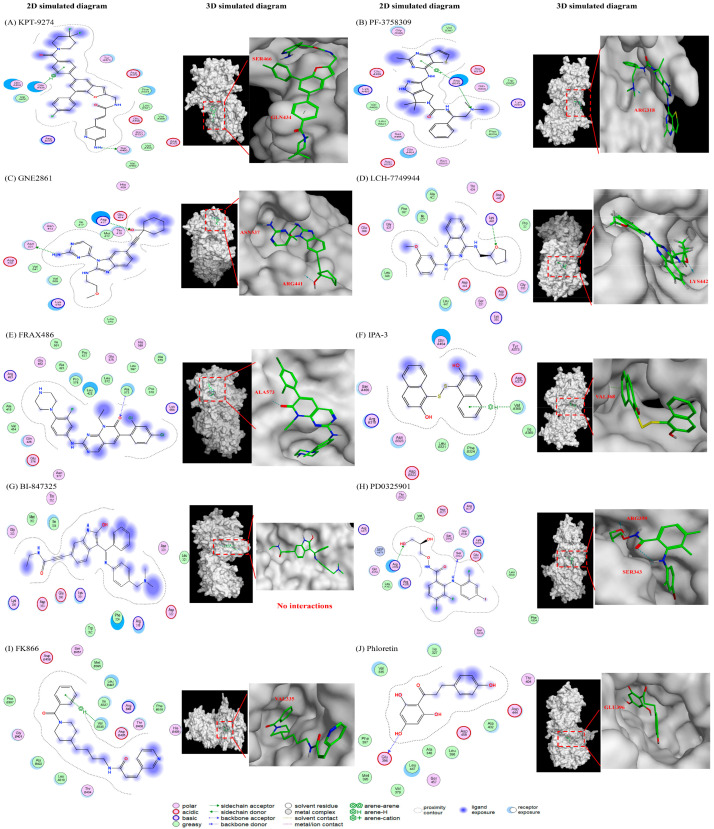
The predicted interactions between different drugs and the PAK4 protein. (1) PAK4 inhibitors: (**A**) KPT-9274, (**B**) PF-3758309, (**C**) GNE-2861, and (**D**) LCH-7749944. (2) PAK1 inhibitors: (**E**) FRAX486 and (**F**) IPA-3. (3) MEK inhibitors: (**G**) BI-847325 and (**H**) PD0325901 (mirdametinib). (4) Positive control (**I**) FK866 and negative control (**J**) phloretin.

**Figure 6 ijms-25-10138-f006:**
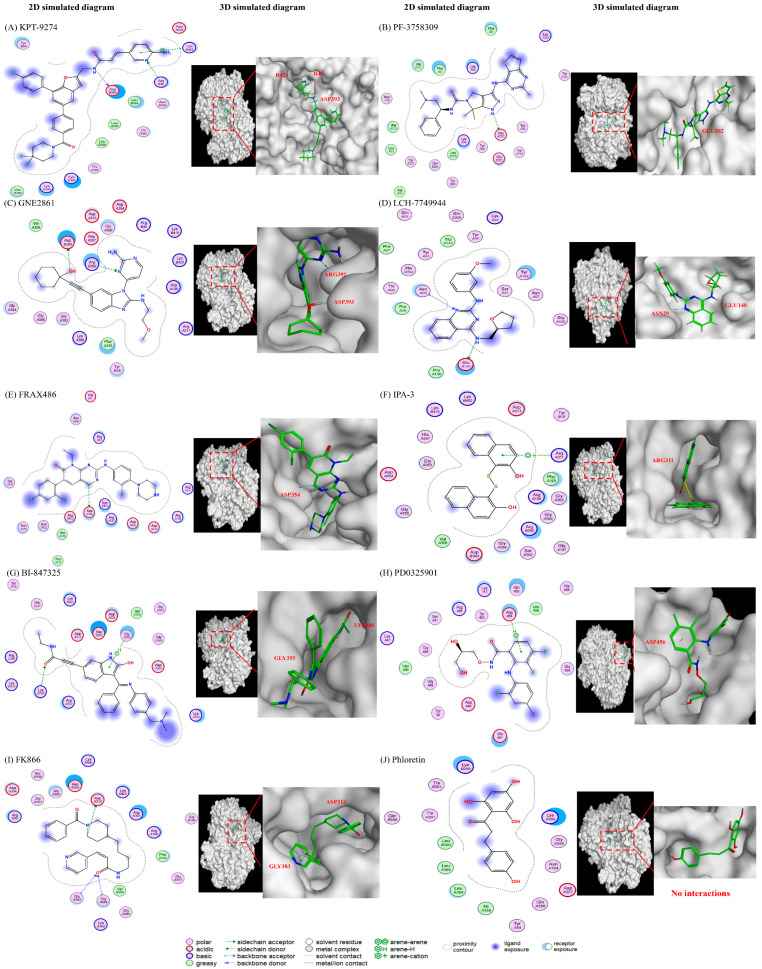
The predicted interactions between different drugs and the NAMPT protein. (1) PAK4 inhibitors: (**A**) KPT-9274, (**B**) PF-3758309, (**C**) GNE-2861, and (**D**) LCH-7749944. (2) PAK1 inhibitors: (**E**) FRAX486 and (**F**) IPA-3. (3) MEK inhibitors: (**G**) BI-847325 and (**H**) PD0325901 (mirdametinib). (4) Positive control (**I**) FK866 and negative control (**J**) phloretin. The results show that PAK4 inhibitors are predicted to interact with PAK4 via pyrimidines and their benzo derivatives. The different positions of nitrogen on the benzene will affect the electrostatic interactions with an -NH bond and may lead to varying levels of NAMPT inhibition. PAK1 inhibitors (IPA-3) do not have the same groups, which may explain why they are less efficient at inhibiting NAMPT.

**Table 1 ijms-25-10138-t001:** Ligand interactions report on PAK4 drugs with PAK4 protein.

No.	Drugs [a]	Score (kcal/mol) [b]	Ligand	Receptor		Interaction	Distance	E (kcal/mol)
1	KPT-9274	−7.5	N10	OG	SER466(A)	H-donor	2.97	−1.6
			6-ring	CG	GLN434(A)	pi-H	3.97	−0.8
2	PF-3758309	−7.1	N8	NH1	ARG318(B)	H-accepter	3.46	−2.2
			6-ring	NH1	ARG318(B)	Pi-cation	3.67	−0.7
3	GNE2861	−6.37	N8	OD1	ASN537(A)	H-donor	3.29	−1.4
			O1	NH1	ARG411(A)	H-accepter	3.43	−0.9
4	LCH-7749944	−6.41	O1	NZ	LYS442(A)	H-acceptor	3.17	−1.6
5	FRAX486	−6.23	O4	CA	ALA573(B)	H-acceptor	3.51	−0.5
			O1	NH1	ARG411(A)	H-acceptor	3.43	−0.9
6	IPA-3	−6.15	6-ring	CG1	VAL368(A)	pi-H	3.58	−0.5
7	BI-847325	No interactions						
8	PD0325901 (mirdametinib)	−6.5	N9	O	SER343 (B)	H-donor	2.90	−3.1
			O7	NH2	ARG359(A)	H-accepter	3.02	−3.1
9	FK866	−6.17	6-ring	CG1	VAL335(B)	pi-H	3.96	−0.5
10	Phloretin	−5.3	O4	O	GLU396 (B)	H-donor	2.91	−3.9

[a] The inhibitors used in the study. [b] Docking Score between inhibitors and PAK4. A lower docking score indicates stronger binding affinities between a protein and a drug.

**Table 2 ijms-25-10138-t002:** Ligand interactions report on PAK4 drugs with NAMPT protein.

No.	Drugs [a]	Score (kcal/mol) [b]	Ligand	Receptor		Interaction	Distance	E (kcal/mol)
1	KPT-9274	−7.25	N8	O	ASP393(B)	H-donor	3.37	−0.5
			N9	NH1	ARG40(B)	H-accepter	3.22	−1.7
			N9	NH1	ARG40(B)	H-accepter	3.07	−3.4
			6-ring	CE	LYS423(B)	Pi-H	3.72	−0.5
2	PF-3758309	−7.2	N4	OE1	GLU202(A)	H-donor	3.01	−1.5
3	GNE2861	−7.8	O1	OD2	ASP393(B)	H-donor	2.84	−3.6
			N6	NH2	ARG392(B)	H-acceptor	3.07	−0.8
4	LCH-7749944	−6.6	N3	OE1	GLU140(A)	H-donor	3.00	−3.0
			N5	N	ASN29(A)	H-acceptor	3.21	−1.1
5	FRAX486	−6.4	C29	OD2	ASP354(A)	H-donor	3.36	−0.6
6	IPA-3	−6.6	6-ring	NH1	ARG311(A)	Pi-cation	4.06	−0.8
7	BI-847325	−6.0	O2	NZ	LYS400(B)	H-acceptor	3.05	−1.2
			5-ring	N	GLY355(A)	pi-H	3.39	−0.6
8	PD0325901 (mirdametinib)	−6.42	6-ring	CB	ASP456(A)	pi-H	3.61	−0.5
9	FK866	−6.9	C10	OD1	ASP313(A)	H-donor	3.41	−0.5
			O2	CA	GLY383(A)	H-accepter	3.24	−0.5
			O2	N	GLY384(A)	H-accepter	3.03	−1.2
10	Phloretin	No interactions						

[a] The inhibitors used in the study. [b] Docking Score between inhibitors and NAMPT. A lower docking score indicates stronger binding affinities between a protein and a drug.

## Data Availability

The data used to support the findings of this study are available from the corresponding authors upon request.

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
