# Peer review of "Inhibition of NAMPT by PAK4 Inhibitors"

_ijms, 2024, doi:10.3390/ijms251810138_

Round 1

Reviewer 1 Report

Comments and Suggestions for Authors

The author evaluated multiple PAK4 inhibitors in biochemical and cellular assays to determine their ability in NAMPT inhibition, which is an off-target effect that has been reported for one PAK4 inhibitor KPT-9274. The author found that in addition to KPT-9274, several PAK inhibitors also inhibit NAMPT activity based on biochemical and cellular assays. Finally, the author tried to provide insight into structural-activity relationships based on computational modeling. Below are my specific critics:

1. What's the rationale for choosing the concentrations? The author used 0.4 mM FK866 in Fig 1 but 100 nM in Fig 2; Why was KPT-9274 used at 100 and 300 nM but PF-3758309 used at 100 nM and 5 uM? Other inhibitors were also used at a single or two concentrations without a pilot study. 

2. In Fig 3A, PF-3758309 at 5 uM has less inhibition than 100 nM, was this as expected? Why?

3. Does the computational analysis match the experimental result? GNE2861 showed the strongest binding in Table 2, Is it the most potent one based on the experiments?

4. The bar graphs are difficult to read. Please consider grouping the same compound together or try a different approach. 

5. It would be better to provide individual values on the bar graph instead of only showing the error bar.

6. I recommend illustrating the inhibition rate normalized with the control group (Similar to Fig 1A) instead of showing the original reading values (Fig 3A)

Author Response

Dear Reviewer 1,

Thank you for your constructive suggestions. We have made several changes based on your suggestions and we believe these changes have improved the paper. Please see the responses to your critiques below. 

  1. Reviewer 1 asked about the basis for the concentrations that were used for the different inhibitors.

       This is an important point because each inhibitor is different, and we wanted to use the appropriate concentration for each one.

          For all the inhibitors used in our study, we used the following to assess the concentrations to use in our assays. FK866 was the positive control for NAMPT inhibition. It was used at the manufacturer's (CycLex Co., Ltd.) recommended concentration in the in vitro assays, and in cells. It was used at concentrations that were shown to inhibit NAMPT in published papers. 

          For PAK4 inhibitors, the important thing was to use a concentration that inhibited PAK4 in previous studies. We used these concentrations because we wanted to demonstrate whether a concentration that could inhibit PAK4, could also inhibit NAMPT. For KPT-9274 the concentrations used were based on our own previous work, and other published papers demonstrating what concentration of the inhibitor was found to inhibit PAK4 and NAMPT. For all other PAK4 inhibitors, we used a combination of factors to determine what concentrations to use. Most importantly, we assessed what concentrations other investigators found to be effective at inhibiting PAK4. We also used concentrations that other investigators found to have an effect on cell growth for the cell-based assays. It should also be noted that for consistency, all PAK4 inhibitors were also tested at 100 uM (cell free assays) and 100 nM (in cells). The only exception was LCH-7749944 which was used at a higher concentration in cells based on other published studies  [1-13].

Point 2) The reviewer asks why PF-3758309 showed a lower inhibition of NAMPT when used at a higher concentration.

              This is an important question and one that was surprising to us as well. We brought this up in the discussion section. Please note that the experiment referred to was done in cells, while the first figure was an in vitro cell free assay. We expect that the result from cell studies may be due to the more complex environment in cells compared with cell free assays.  We want to highlight that the most important message from this result is that overall our NADP studies validate our cell free studies, and our cell based NAD+ assays.

This was addressed in the discussion section of the paper.

Point 3) The reviewer asks whether the computational analysis matches the experimental results.

               The computational analysis is highly effective as a model for predicting the mechanism by which PAK4 inhibitors bind to and inhibit target proteins. However, these types of modeling studies do not take the place of validation in the lab. Our computational analysis showed that PAK4 inhibitors can have a strong affinity with NAMPT, which was supported by the experiments (Fig 1, and Fig 2). Further studies in the lab would be required to determine which one is quantitatively strongest. 

Points 4 and 5) The reviewer indicates that the bar graphs were difficult to read and that individual values would be helpful.

              Due to experiments being done at different times and drug concentrations, we felt it was more straightforward to present the results as separate graphs. However, presenting individual values is an excellent suggestion. We have provided individual values on the graph for Figure 1 and Figure 2 in the paper. Figure 3 was quite large and the graph appeared messy with individual values, but we attached the graph with individual values here.

Figure 3 with individual values is in the attachment.

Point 6) The reviewer suggested that we present some of the data differently.

We noticed that NAD/NADP results are usually presented in terms of RLU levels and feel this is the most clear and consistent way to present the data. However, we think that our results are now clearer since we added the percentage of NAD/NADH levels. We also added an explanation about the values of the bar graph in Figure 2.

References:

  1. Pittelli M., et al. J Biol Chem 2010, 285, 34106-34114, doi:10.107/jbc.M110.136739.
  2. Rane C., et al. A.Sci Rep 2017, 7, 42555, doi:10.1038/srep42555.
  3. Staben S.T., et al. J Med Chem 2014, 57, 1033-1045, doi:10.1021/jm401768t.
  4. Wang Y., et al. PLoS One 2016, 11, e0153312, doi:10.1371/journal.pone.0153312.
  5. Murray B.W, et al. Proc Natl Acad Sci U S A 2010, 107, 9446-9451, doi:10.1073/ pnas.0911863107.
  6. Deacon S.W., et al. Chem Biol 2008, 15, 322-331, doi:10.1016/j. chembiol.2008.03. 005.
  7. Kim J., et al. Nutrients 2018, 10, doi:10.3390/nu10070868.
  8. Chang W.T., et al. E Food Chem 2012, 134, 972-979, doi:10.1016/j.foodchem.2012. 03.002.
  9. Samimi H., et al. Cancer Cell Int 2022, 22, 388, doi:10.1186/s12935-022-02813-6.
  10. El-Kishky A.H.M., et al. Med Oncol 2022, 39, 144, doi:10.1007/s12032-022-01738-4.
  11. Van Dort M.E., et al. Bioorg Med Chem 2015, 23, 1386-1394, doi:10.1016/j.bmc.201 5.02.053.
  12. Zhang J., et al.Cancer Lett 2012, 317, 24-32, doi:10.1016/j.canlet.2011.11.007.
  13. Ramos-Alvarez I., et al. Am J Physiol Gastrointest Liver Physiol 2018, 315, G302-G317, doi:10.1152/ajpgi.00005.2018.

Reviewer 2 Report

Comments and Suggestions for Authors

Comments on the review of the article "Inhibition of NAMPT by PAK4 Inhibitors in Triple-NegativeBreat Cancer"

This study, at the in vitro level, demonstrated that PAK 4 inhibitors inhibit not only PAK 4 activity but also NAMPT activity and result in reduced NAD and NADP levels in cells, as a role to inhibit the growth of cancer cells. The research is novel, which is a further of the preliminary work of the research team, and has certain academic value. However, the authors need to further answer the following questions:

1. Why did not design in vivo experiments, such as the construction of breast cancer model in nude mice, and then observe the mechanism;

2. What is the basis for the concentration of PAK 4 inhibitors and the intervention time point selected in this study? Are consistent with the in vivo experiments.

3. In the summary of this study, the main purpose, method and results should be written; the preface is too much to be further refined.

Comments on the Quality of English Language

NO

Author Response

Dear Reviewer 2,

Thank you for your constructive suggestions. We have made several changes based on your suggestions and we believe these changes have improved the paper. Please see the responses to your critiques below. 

  1. The reviewer asked why we did not design in vivo experiments, such as the construction of breast cancer model in nude mice, and then observe the mechanism.

While this is an excellent suggestion for future studies, the current work was designed only to understand the mechanism of action of the Pak4 inhibitors. PAK4 inhibitors have already been shown to inhibit the growth of several types of cancer cells. The mechanism by which it functions however is still unclear, and this basic mechanism is what we wanted to address here. Our results open the door for more studies in the future where the mechanism of action of Pak4 inhibitors in vivo can be explored. The current study is designed only to test what kinds of targets these drugs inhibit.

  1. The reviewer asked about the concentrations we used for the different inhibitors.

      This is an important point because each inhibitor is different, and we wanted to use the appropriate concentration for each one.

          For all the inhibitors used in our study, we used the following to assess the concentrations to use in our assays. FK866 was the positive control for NAMPT inhibition. It was used at the manufacturer's (CycLex Co., Ltd.) recommended concentration in the in vitro assays, and in cells. It was used at concentrations that were shown to inhibit NAMPT in published papers. 

         For PAK4 inhibitors, the important thing was to use a concentration that inhibited PAK4 in previous studies. We used these concentrations because we wanted to demonstrate whether a concentration that could inhibit PAK4, could also inhibit NAMPT. For KPT-9274 the concentrations used were based on our own previous work, and other published papers demonstrating what concentration of the inhibitor was found to inhibit PAK4 and NAMPT. For all other PAK4 inhibitors, we used a combination of factors to determine what concentrations to use. Most importantly, we assessed what concentrations other investigators found to be effective at inhibiting PAK4. We also used concentrations that other investigators found to have an effect on cell growth for the cell-based assays. It should also be noted that for consistency, all PAK4 inhibitors were also tested at 100 uM (cell free assays) and 100 nM (in cells). The only exception was LCH-7749944 which was used at a higher concentration in cells based on other published studies  [1-13].

  1. The reviewer suggests that in the summary of this study, the main purpose, method, and results should be re-written and clearly defined.

We agree, and have revised the summary paragraph., Furthermore, the entire text has been re-written and we have made sure to make our main points clear and concise.

References:

  1. Pittelli M.,et al. J Biol Chem 2010, 285, 34106-34114, doi:10.1074/jbc.M110.136739.
  2. Rane C., et al., A.Sci Rep 2017, 7, 42555, doi:10.1038/srep42555.
  3. Staben S.T.,et al. J Med Chem 2014, 57, 1033-1045, doi:10.1021/jm401768t.
  4. Wang Y., et al. PLoS One 2016, 11, e0153312, doi:10.1371/journal.pone.0153312.
  5. Murray B.W.,et al. Proc Natl Acad Sci U S A 2010, 107, 9446-9451, doi:10.1073/pnas. 0911863107.
  6. Deacon S.W., et al. Chem Biol 2008, 15, 322-331, doi:10.1016/j. chembiol.2008.03. 005.
  7. Kim J.,et al. Nutrients 2018, 10, doi:10.3390/nu10070868.
  8. Chang W.T., et al. E Food Chem 2012, 134, 972-979, doi:10.1016/j. foodchem.2012. 03.002.
  9. Samimi H., et al. Cancer Cell Int 2022, 22, 388, doi:10.1186/s12935-022-02813-6.
  10. El-Kishky A.H.M., et al. Med Oncol 2022, 39, 144, doi:10.1007/s12032-022-01738-4.
  11. Van Dort M.E., et al. Bioorg Med Chem 2015, 23, 1386-1394, doi:10.1016/ j.bmc.201 5.02.053.
  12. Zhang J., et al.Cancer Lett 2012, 317, 24-32, doi:10.1016/j.canlet.2011.11.007.
  13. Ramos-Alvarez I., et al.. Am J Physiol Gastrointest Liver Physiol 2018, 315, G302-G317, doi:10.1152/ajpgi.00005.2018.

Reviewer 3 Report

Comments and Suggestions for Authors

The manuscript entitled Inhibition of NAMPT by PAK4 inhibitors in Triple-Negative Breast Cancer by Wand and Minden describes, using both in vitro and in cellulo assays the inhibition of these two targets by different kinds of inhibitors. These parameters were evaluated in the aim to deciphering whether the dual inhibition mechanism could be at the source of compounds antitumoral efficacy.

Although the intentions are noble, the means by which the authors opted to perform the assays and to present their data are not appropriate and provide sufficient flaws to prevent them from publication at the current stage. The manuscript is written in a format that somehow looks partially like a review paper with exhaustive repetition between section which strongly dilute the message that they try to vehicle.

The manuscript structure and format is deficient in several points, which I attempted to summarize here:

Manuscript writing and format:

·         The title is not stating what was done and sounds like a review title. The use of a single cancer cells cannot be rounded to TNBC and should instead be considered with simply mentioning the TNBC cell line

·         Abstract; the results and findings and how these were done are not there, again general review-like statements like ‘’ raise important questions about …’’ are used.

·         Some paragraph use PAK4 whereas others use Pak4 (like if sentences were organized together from different sources/writers. This must be arranged.

·         The histograms are very difficult to evaluate in the way they are presented. As comments below will, however, highlight, these data should be curves instead, ideally with calculated values like ideally Ki or IC50 to compare them.

Introduction:

·         Lines 40-42, it is mandatory to state what kind of degree of evidence in the literature that is supportive about these ‘’roles’’ to make readers understand e.g. silencing/overexpression-inhibition, in vitro, in cellulo, in vivo studies, human specimens… The same line of thinking should be used for the recognition as a ‘’promising target’’ to properly position the pharmacology behind the target.

·         Difficult to treat twice in 4 lines (lines 68-71)

·         Lines 42-47, the organ distribution is not bringing anything.

·         In the context of the study, it would be appropriate to contextualize the background behind KPT-9274 and the known off-target it can have as this is the central theme of this manuscript.

·         Lines 82-84; The segment about NAD+ that ‘’improve health and boosts immunity’’ should be toned down, this is meant to be a pharmacology paper after all.

Results:

·         Figures are not used in their numerical order (Line 125; Figures 1 and 4)

·         The first paragraph is massive to assimilate. A Table would make it much easier to follow. The table could have compounds name, structures, PAK4 and/or other PAK inhibition (ideally constants, or at worst subjective ordinal values like +++/++/+/-) along with references, ideally potency (constants, Ki, IC50) would be used, whenever applicable.

·         All figures have a caption below, but there are also figure legends at the end of the manuscript. It is therefore confusing which is the proper one.

·         The authors tested against SUM159 cells, but the rationale of using them is not present. Why them and not other one for example. This is important, especially given the fact that they only tested on these cells and no other one.

Discussion:

·         Paragraphs in lines 245-267 and 292-313 are mostly repeating what was in introduction and in results, sometimes almost using the same wording.

·         Figure 7 is really inappropriate and could be a simple panel with ? on every arrow, as nothing there is verified.

Yet, the most important concerns are the following:

·         Pharmacology/Affinity: when comparing compounds inhibitory effect against enzyme in an in vitro assay or potency in a given phenotype in cell-based assay, pharmacology warrant the use of dose-response to properly compare them. The authors have performed their assays with very different doses across the compounds. The selection of these doses and the rationale behind them is not present and is critical. In fact, comparing compounds like this for their inhibitory capacity requires the use of different concentrations to draw concentrations-response curves, not curve with Absorbance values (like Figure 1C-D). Only then can the compound inhibition be compared against their target. Why are Figure 1A and B distinct but testing the same compounds, sometimes with different doses? There is even a reference to Figure 1F (Line 144) which does not exist.

·         The authors opted to measure cell-based NAD/NADH and NADP/NADPH (it is not so clear what the second measurement is bringing) after exposure to compound for 6h or 24h. They state at numerous places that this is tricky as they also observe effect on cell survival and present MTS assay data as cell survival values alongside their results. MTS assay measure cell viability by directly measuring the reduction of the reagent into a colored formazan by NAD/NADH itself. It is therefore not the best procedure to evaluate cell survival in this context (actually it redundantly assesses the levels of NAD/NADH, so it is not surprising that both Figure 2A and 2B are so similar). Given the very short exposure period (i.e. 6h) observing mortality by 60% or more is completely aberrant and illustrate that the dose have not been properly selected (and are way too much cytotoxic) but it most likely reflect NAD/NADH instead. Of course, all these comments apply to NADP/NADPH data as well.

·         An important point missing is the lack of dose-response curves between compounds to compare their potency in these assays, along with the impossibility to assess whether NAD/NADH can really be a proxy of PAK4 target engagement in cells.

·         The statement on Line 204 saying that Results indicated that the PAK4 inhibitors interact with NAMPT appear farfetched. Can we really confirm this solely on this in silico approach?

·         Since NAD/NADH balance in cells is affected by a lot of mechanisms, proper target engagement evaluation strategies would be needed to assess whether the statement from the authors are actually right about the dual-inhibition in cellular context. The use of cellular thermal-shift assay directly against the target would be needed to claim this. Therefore, the general tone of the findings, if found true by comparing compound potency using different concentrations, would still need to be toned down.

·         Statistical Analysis: In material and methods, there is a section about statistics but, apart from few statement like in Line 131 (significant), it does not look like the statistics were used.

For all these reasons, I feel like simply asking for major revision is not enough because the portion of the manuscript simply needing revision is too small compared to the parts needing complete revision/experimentations.

Comments on the Quality of English Language

Typos remaining, a lot of repetition (see section above)

Author Response

Dear Reviewer 3,

Thank you for your constructive suggestions. We have made several changes based on your suggestions and we believe these changes have improved the paper. Please see the responses to your critiques below. 

Point 1:

We agree with the reviewer that the title does not accurately reflect the main point of our paper. Our main premise is that Pak4 inhibitors can inhibit NAMPT, and that this is important for considering how PAK4 inhibitors may inhibit the growth of cancer cells. We change the title to: The inhibition of NAMPT by PAK4 inhibitors.

Point 2.

      Thank you for your suggestions about the abstract. We have re-written the abstract to make it more focused on the study, methods, and findings.

Point 3: We have gone over the manuscript carefully and made sure to be consistent as to how PAK4 was written.·       

      Point 4: We have re-made some of the graphs by including percentages to make the results more clear. The current study is a small study and currently we did not carry out concentration curves. We explain how we decided on the concentrations of the drugs we used below (question 11).

Point 5: The reviewer suggests several changes in the introduction. We have added several more details about indicating why Pak4 is a promising target and what roles it has in cellular functions such as proliferation.

Point 5:

This is a combined answer to several questions about the text. We have fixed several errors in the text. These include a sentence with repetitive words, and we have made sure that the figures are now all properly referred to, and in their correct order. We have re-written the first paragraph the result section, as the author indicates there is too much information here. We have also fixed the figure legends, and moved them to below the figures, so that they are only seen in one part of the paper.

Point 6.

       As the author suggested we have removed the discussion of organ distribution. Regarding off-targets, no major direct KPT-9274 off-targets have been reported but have focused on Pak4 and NAMPT.

       Point 7.

      There are several areas where the reviewer suggests that the overall tone of the paper need to be toned down. We agree, and have revised the paper accordingly.

      Point 8:

      We agree that we did not make it clear why we tested SUM159 cells. We have used this cell line in previous studies of Pak4 inhibition by KPT-9274. However, the main point of the cell-based assays was to validate our in vitro cell free assays in Figure 1. We found that in fact, the results in cells (NAD+ and NADPH levels) are consistent with our cell-free results. We agree that using one cell line is not sufficient to make a general statement about TNBC, however, and have changed our title accordingly, as indicated above.

Point 9:

We have largely re-written the discussion section to make it more concise and to avoid repetitive wording.

Point 10. We have re-made Figure 7.

Point 11.

This point is in response to several questions that the reviewer had about drug concentrations. The reviewer is correct that we tested only a few concentrations of the inhibitors used here and that a dose-response curve would have advantages. However, we chose the doses we used for specific reasons, as indicated below.

For all the inhibitors used in our study, we used the following to assess the concentrations to use in our assays. FK866 was the positive control for NAMPT inhibition. It was used at the manufacturer's (CycLex Co., Ltd.) recommended concentration in the in vitro assays, and in cells. It was used at concentrations that were shown to inhibit NAMPT in published papers. 

      For PAK4 inhibitors, the important thing was to use a concentration that inhibited PAK4 in previous studies. We used these concentrations because we wanted to demonstrate whether a concentration that could inhibit PAK4, could also inhibit NAMPT. For KPT-9274 the concentrations used were based on our own previous work, and other published papers demonstrating what concentration of the inhibitor was found to inhibit PAK4 and NAMPT. For all other PAK4 inhibitors, we used a combination of factors to determine what concentrations to use. Most importantly, we assessed what concentrations other investigators found to be effective at inhibiting PAK4. We also used concentrations that other investigators found to have an effect on cell growth for the cell-based assays. It should also be noted that for consistency, all PAK4 inhibitors were also tested at 100 uM (cell free assays) and 100 nM (in cells). The only exception was LCH-7749944 which was used at a higher concentration in cells based on other published studies  [1-13].

      Point 12.

       Regarding Figure 1, we made two distinct figures because these were independent experiments, in some cases testing new doses and new drugs that became available.

      Point 13.

      We agree that testing NAD and NADP is in some ways redundant. However, these separate assays were carried out to validate the consistency of the results. The MTS assay was carried out only to determine whether the cells were surviving at the doses used. In addition, while it is true that some doses led to a decrease in cell viability, making the interpretation difficult, the most important message about the cell-based studies, however, is to show that the NAD and NADP assays are consistent with what was seen in cell-free assays.

      Point 14:

      The reviewer is correct that we cannot confirm an interaction solely based on our MOE data. This is instead a prediction. The main purpose here is to provide a virtual model that may help us better understand the potential interactions. This is addressed in the discussion.

      Point 15.

      We agree with the need to tone down some of our writing, based on the evidence that we present. We have re-written the paper carefully and taken care to correct grammatical errors.

Point 16.

Statistical analysis:

For the Statistical analysis, all data were presented as mean ± standard deviation (STD), with error bars reflecting this variability. An independent Student's t-test was employed to compare the means of two groups. Any two groups referred to as significantly different in the text had a P<0.05. This was clarified in the materials and methods section.

References:

  1. Pittelli M., et al. J Biol Chem 2010, 285, 34106-34114, doi:10.107/jbc.M110.136739.
  2. Rane C., et al. A.Sci Rep 2017, 7, 42555, doi:10.1038/srep42555.
  3. Staben S.T., et al. J Med Chem 2014, 57, 1033-1045, doi:10.1021/jm401768t.
  4. Wang Y., et al. PLoS One 2016, 11, e0153312, doi:10.1371/journal.pone.0153312.
  5. Murray B.W, et al. Proc Natl Acad Sci U S A 2010, 107, 9446-9451, doi:10.1073/ pnas.0911863107.
  6. Deacon S.W., et al. Chem Biol 2008, 15, 322-331, doi:10.1016/j. chembiol.2008.03. 005.
  7. Kim J., et al. Nutrients 2018, 10, doi:10.3390/nu10070868.
  8. Chang W.T., et al. E Food Chem 2012, 134, 972-979, doi:10.1016/j.foodchem.2012. 03.002.
  9. Samimi H., et al. Cancer Cell Int 2022, 22, 388, doi:10.1186/s12935-022-02813-6.
  10. El-Kishky A.H.M., et al. Med Oncol 2022, 39, 144, doi:10.1007/s12032-022-01738-4.
  11. Van Dort M.E., et al. Bioorg Med Chem 2015, 23, 1386-1394, doi:10.1016/j.bmc.201 5.02.053.
  12. Zhang J., et al.Cancer Lett 2012, 317, 24-32, doi:10.1016/j.canlet.2011.11.007.
  13. Ramos-Alvarez I., et al. Am J Physiol Gastrointest Liver Physiol 2018, 315, G302-G317, doi:10.1152/ajpgi.00005.2018.

Round 2

Reviewer 1 Report

Comments and Suggestions for Authors

My questions have been addressed. 

Author Response

Dear Reviewer 1,

Thank you so much for your valuable comments. We are glad that you are satisfied with our responses.

Best regards,

Yiling Wang

Reviewer 3 Report

Comments and Suggestions for Authors

Although the authors did correct the structure of the manuscript. I still disagree with the data and ways they are bringing forward some of the conclusions. Even if they mention that this is a ''small study'', this does not make data interpretation short-cutting OK. To me this should be conducted completely differently to ensure the conclusions stands with the data.

Figure 1- The lack of dose response and the way the compounds concentrations were tested is puzzling. Yes KPT-9274 has stronger inhibition at 10 µM than 100 µM, as they point out, but for most compound the low and higher dosing give the same levels of inhibition (1-5 vs 100 µM). 

Still, the authors use terminology like ''weaker/strong/strongest'' and compare to their positive control (at 0.4 mM!) Only curve, or having reasons to beleive that we have a dynamic assay here would allows such comparisons of potency to be made. If the concentrations used were explained, maybe we could follow how this was methodologically designed ? Here its not the case. 

Figures 2-3. Again, dose selection is confounding. Although we again do not observe dose-response (likely because the assay is run over too long ?) Compounds like PF3758309 show less inhibition at 5 µM than at 100 nM (50 times more !). Moreover, comparison of potencies are being made although the doses are quite spaced. e.g. between FRAX-486 and LCH-7749944 at respectively 500 nM and 10 or 30 µM !. 

Although the NAD/NADH or NADP/NADPH data are likely useable, they simply can't be looked at using a MTS viability assay to evaluate if the cells are alive or death. The MTS is pretty much a way to also measure NAD/NADH or NADP/NADPH by evaluating metabolic conversion, it cannot serve as a reference (and as we observe, curiously NAD/NADH and NADP/NADPH data are almost directly matching the MTS assay, as we would expect for this specific reason). Any reference to cell viability/mortality here should have been done using another type of measurements (e.g. LDH release, cell count, etc...) I would thus demand that they are removed or experiment done again. 

Figure 7 really is not bringing anything. I beleive any reader can understand what is meant here by a few of the discussion sentences.

Author Response

Dear Reviewer,

Thank you so much for your comments. Please see our responses in the attachment.
